# Multi-channel portable odor delivery device for self-administered and rapid smell testing
Richard Hopper [1,2] ✉, Daniel Popa [2] ✉, Emanuela Maggioni [1,3] ✉, Devarsh Patel[1], Marianna Obrist[3], Basile Nicolas Landis [4], Julien Wen Hsieh[4] & Florin Udrea[2]

To improve our understanding of the perception of odors, researchers are often required to undertake experimental procedures with users exposed to multiple odors in a variety of settings, including to diagnose smell loss in clinics and care homes. Existing smell tests are typically administered using multiple sniffing pens, manually presented to patients by a highly specialized nurse using a time-consuming and complex testing paradigm. Automated odor delivery devices, such as olfactometer systems, exist but are expensive, bulky and typically lab based, making them difficult to use for on the ground odor delivery. Here we have developed a portable, affordable, odor delivery device that can deliver 24 odors through individual channels with high temporal precision and without cross-contamination. The device allows for the rapid, flexible sequencing of odors via digital control using a mobile application and has been experimentally validated in the lab, as well as tested on patients. The design provides several advantages for investigating olfactory perception and offers the possibility that users can one day self-administer smell tests in a range of settings, including at home, allowing smell healthcare services to evolve and become part of a routine practice and self-care culture.

The sense of smell is one of our five main senses that links us to the world around us and plays a vital role in our health and well-being. Just like with other senses, any disruption or loss to our ability to smell (i.e., smell dysfunction), can have a debilitating impact on our quality of life, including affecting our emotional, cognitive, and/or mental health[1,2]. For example, smell dysfunction can reduce social confidence because of the inability to reliably assess personal hygiene state and carries an increased risk to well-being and personal safety (e.g., the inability to judge food safety, detect fire hazards, leaking gas, etc.)[3]. Distortions in olfactory perception usually have a profound impact on the perception of food and drink flavors[4], which are multi-sensory percepts, combining input from gustatory and olfactory perception among others. A reduced ability to enjoy food can have a negative effect on nutrition, and/or the immune system[5].

Smell dysfunction is also an important biomarker for various neurological diseases[6]. For example, it is well established that olfactory impairment predicts incident mild cognitive impairment and progression to Alzheimer's disease[7,8]. With life expectancy rising, there are an increasing number of age-related neurodegenerative diseases (like Alzheimer's[7] and Parkinson's[9]), contributing to increased pressure on healthcare providers. There is, therefore, a growing need for innovation to facilitate the introduction of

personalized and stratified medicine, with a focus on the early diagnosis of disease, prevention rather than cure, care closer to the home, and continuous health monitoring, rather than periodic sampling[3,10]. This calls for a more engaged public with higher levels of awareness of smell health and care that will assist in relieving the financial pressure of such situations by adopting innovative diagnostic technology, allowing them to remain healthier and independent for longer[11].

Tests to evaluate our sense of smell are complicated by a number of factors, including the high dimensionality of the olfactory stimulus space and the large dynamic range of human smell receptors. Overcoming these challenges often requires testing with a large number of odorants of different types and dilutions[12] to adequately cover even a portion of the olfactory stimulus space. Smell tests also need to be rapid and easy to administer, without sacrificing the quality of clinically important data, such as olfactory threshold values[13].

By analogy to hearing tests, which measure the lowest perceived intensity of a sound, olfactory threshold tests measure the lowest perceived concentration of an odor. Existing threshold tests use multiple felt-tip pens filled with serial odorant dilutions[14], which are manually presented to the patient by a highly specialized nurse using a time-consuming and complex

[1]OWidgets Ltd. - OW Smell Made Digital, London, E1 1DU, UK. [2]Department of Engineering, University of Cambridge, Cambridge, CB3 0FA, UK. [3]Department of Computer Science, University College London, London, NW1 2AE, UK. [4]Rhinology-Olfactory Unit, Department of Otorhinolaryngology-Head and Neck Surgery, Geneva University Hospitals, Geneva, 1205, Switzerland. ✉e-mail: richard@owidgets.co.uk; dp387@cam.ac.uk; emanuela@owidgets.co.uk

testing paradigm[15]. Although the olfactory threshold test is a clinically essential component of smell evaluation, this test is rarely performed in clinical settings because of its complex and lengthy procedure[16].

Olfactometers are widely used throughout olfactory research, enabling the automated delivery of temporally controlled odor stimuli to subjects. Olfactometer systems typically have a number of common components, including a filtered air supply delivered by a pump from the environment[17] or from a gas cylinder[18]; one or more temporally activated odor sources[19]; delivery channels; and a method of spatially directing odor to the user, such as through a face mask[18,20] or nasal attachment[21]. A summary of different approaches for odor delivery is given in Table 1.

Olfactometer systems can generate odorized air in a number of ways, including through the use of bottles containing liquid odorant[17]; gas sample bags[22]; gas-filled syringes[23]; and active thermal evaporation of liquid odorant using a heated plate[24].

To determine a patient's odor threshold, the olfactometer system must be able to generate a range of odor concentrations. The odor concentration can be controlled by diluting the odor stream with clean air[22], however, gaseous dilution typically requires the use of proportional valves to control the dilution ratio which can add cost and complexity to the system. In addition, the use of a common mixing chamber can lead to cross-contamination between odors, unless the system is carefully cleaned and flushed out between tests[25]. An alternative approach for varying the odor concentration, which avoids cross-contamination and expense, is the use of multiple channels with serially diluted odorants[19]. With this approach, the airflow through the individual odor channels can be conveniently controlled using an array of solenoid valves and associated electronic control circuitry.

Given the high number of sub-components used in typical olfactometer systems, including pumps; temperature-controlled odorant reservoirs; tubing; valves; and mass flow controllers, olfactometer systems are generally high cost, bulky and limited to lab based environments. Some attempts have been made to develop miniaturized odor delivery devices, for example[18], reported on a miniaturized single channel odor delivery device that uses interchangeable cartridges filled with odor vapor. Miniaturized odor delivery devices have also been commercially developed, including by Aromajoin Corp. (Japan), whose system employs replaceable odorant cartridges activated by piezoelectric air pumps. OVR Tech LLC (US) have also developed a wearable system based on a VR headset fitted with odorant cartridges activated by a piezoelectric atomizer. However, such low-cost miniaturized devices are generally aimed at the entertainment market and often have compromised performance, including limited odor flow, poor control of odor intensity, poor directivity of the odor stream to the user, poor temporal resolution, contamination issues between odor channels and limited flexibility, e.g., due to the use of proprietary odorant cartridges.

There is therefore a need to create compact systems for odor delivery to enable smell tests that are time efficient (able to deliver tens to hundreds of odorants per test session) and flexible to allow the odorant selection to be easily tailored to suit the needs of the experiment.

Here, we describe a portable multi-channel odor delivery device capable of efficient and flexible odor delivery, suitable for research applications in a variety of settings. The digitally controlled device uses an interchangeable odorant cartridge (24-channels), which can be prepared during the course of an experiment, such as that demonstrated in this work for smell testing.

## Methods
### Device description
The odor delivery device presented here utilizes components common to most olfactometer systems, including a clean air supply, solenoid valves for directing airflow to the selected odor source, and a method to deliver the odor to the user. This device is developed by OWidgets Ltd., a University spin-out, formed off the back of international scientific collaborations, including efforts to advance odor delivery methods for smell testing. A cutaway image showing the system components and a pneumatic diagram are shown in Fig. 1a, b, respectively.

To enable odor transport, the device draws air from the environment using a diaphragm pump (Parker, BTX Connect) with a maximum flow rate of 6 L/min. To remove traces of organic compounds, the air is first filtered using an activated carbon filter (Festo, MS4/D-MINI-LFX). The filtered air is then piped to an aluminum manifold, which helps to smooth the airflow, which is distributed to a bank of 24 solenoid valves (Zanty, SDF-0626L) that can be individually activated to direct airflow into separate odor reservoirs. The airflow rate can be adjusted over a range of 2–6 L min$^{-1}$ using a flow regulator (Festo, GRLA-M5-QS-4-D).

Liquid odorants are absorbed on sponge materials within 24 odor reservoirs which are housed in a removable aluminium cartridge. The odorant cartridge is mounted on metal posts and is clamped into place using a pair of latches, permitting flexible deployment for tests. The cartridge material can be cleaned effectively using a baking soda solution and an oven can also be used to evaporate off any residual odorant. Pneumatic sealing of the odor channels is achieved using rubber O-rings. The large number of channels permits the use of odorants of different concentrations and types. The small headspace of the odor reservoirs allows them to quickly fill with saturated vapor. Upon activation of airflow into an odor reservoir, saturated odor vapor is picked up and piped to an outlet channel through Teflon pipes. The use of individual outlet pipes/channels avoids cross-contamination between odors. The odor flow from the pipes is directed toward a focal point 10-cm distance away from the outlet using a resin printed outlet adapter shown in Fig. 1d. The outlet adapter design could be easily modified for other use cases e.g., to vary the focal point.

The functional blocks of the device's electronic control circuitry are shown in Fig. 1c. System control and communication are enabled by a CPU and Bluetooth module (Raytac, MDBT50Q-1MV2) on an Adafruit Feather nRF52840 Express board, integrating a Low Energy Bluetooth 2.4 GHz transceiver and an ARM Cortex-M4 CPU which acts as a low power controller for the rest of the system.

The solenoid valves are controlled by a serial digital output from the CPU which is routed to shift registers to generate a set of 24 parallel digital outputs which are used to switch higher voltage (12 V) MOSFET driver circuitry. Up to three odor channels can be activated simultaneously. The diaphragm pump is controlled using a pulse-width-modulated control

## Table 1 | Odor delivery system approaches

| Ref | Year | Odor source | Control method | Outlet type | Channels |
|---|---|---|---|---|---|
| 17 | 2010 | Odorant bottle | Solenoid-controlled odor lines with flow control and dilution. | Nose piece | 9 |
| 19 | 2019 | Odorant bottle | Solenoid-controlled odor lines. Open air mixing with carrier stream. | Nose piece | 12 |
| 20 | 2018 | Odorant reservoir | Fan coupled to rotatable odor reservoirs. | Outlet port | 8 |
| 29 | 2002 | Odor filled syringes | Motorized syringes. | Face mask | 1 |
| 30 | 2015 | Odorant bottles | Solenoid-controlled odor lines. | Nose piece | 12 |
| 24 | 2018 | Heated liquid odorant | Solenoid-controlled odor lines. | Outlet port | 3 |
| 22 | 2001 | Sample bags | Solenoid-controlled odor lines with flow control and dilution. | Outlet port | 3 |
| 18 | 2005 | Odorant bottle | Odor injection into carrier stream. | Face mask | 1 |

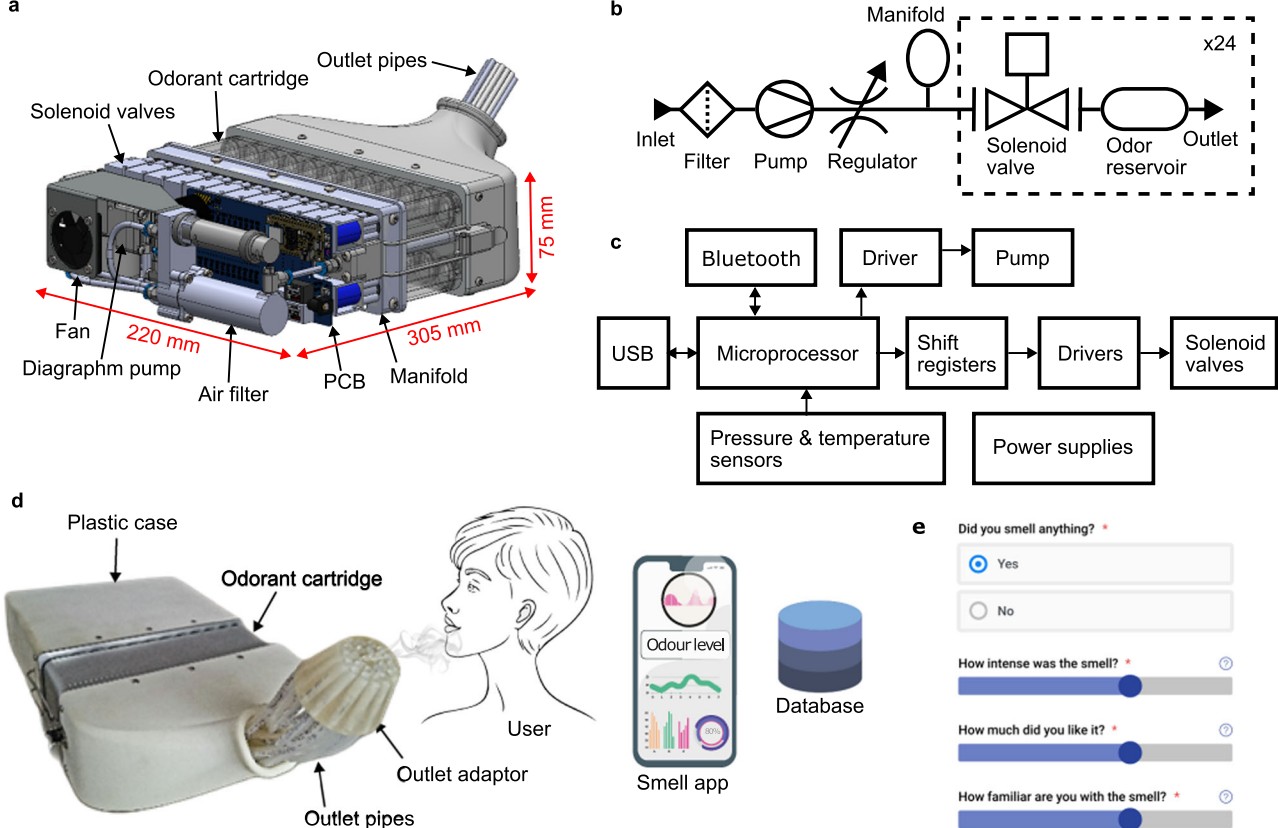

**Fig. 1 | System description. a** Cutaway computer-aided design (CAD) image showing the key components of the 24-channel odor delivery device, including the air filter, diaphragm pump, solenoid valves, printed circuit board (PCB), and odorant cartridge. **b** Pneumatic diagram illustrating the airflow through the components of the odor delivery device. Air from the surrounding environment is filtered, pumped, and channeled by a bank of 24 solenoid valves to an odor reservoir, where it is odorized before reaching an outlet. **c** A system diagram of the electronic control circuitry. A microprocessor module (Raytac, MDBT50Q-1MV2) is used for system control and communication using bluetooth or universal serial bus (USB). Digital lines are interfaced to higher voltage (12 V) drivers for activation of the solenoid valves and diaphragm pump. A pressure sensor is used for diagnostic tests

during system operation. **d** An image of the assembled device and its interaction with the user and mobile application. The device is housed in a three-dimensional (3D) printed plastic case and contains an odorant cartridge which slides onto fixing posts and is screwed into place within the assembly. Individual outlet pipes are directed to the user using an outlet adapter. A mobile app is used for system control and for recording perceptual data from users (stored in a cloud database). **e** A screenshot showing the mobile app used for system control and recording perceptual data from users of the smell delivery device. After each odor exposure, users are asked a series of questions, including inviting them to rate the odor intensity, character, and familiarity. Permission was obtained from Jing Xue (University College London) for the use of the user graphic.

signal from the CPU and is driven using similar driver circuitry, permitting electronic control of the airflow.

Auxiliary components of the electronic circuitry include a pressure sensor (Freescale, MPX 53GP) for monitoring the internal air pressure of the system and a temperature and humidity sensor (Sensirion, SHT21S) for

environmental monitoring. Power to the different sub-modules (3.3/5/12 V rails) is provided by switching regulators from Murata Ltd. Images showing the internals of the device and the odorant cartridge are shown in Supplementary Figs. 1, 2, respectively, and the fully assembled device, housed in its 3D-printed plastic case, is shown in Fig. 1d.

The Bluetooth interface permits mobile control and integration of the device into the Internet of Things (IoT). To facilitate this, a mobile app has been developed for control using JavaScript, which runs on an Android platform. The app can be used for smell testing applications and allows users to easily trigger the odor delivery using a graphical interface and record perceptual feedback. A screenshot of the app is shown in Fig. 1e. After an odor delivery has been triggered, the app presents the user with a questionnaire, allowing them to record their perceptions of odor stimuli. User data recorded during each training session is stored on a cloud server for subsequent analysis.

At current prices, the cost of construction of the odor delivery device totals $2,140, with each odorant tested costing an additional few dollars in disposables (i.e., for the odorant and sponge absorber). The cost of the device is expected to fall if the unit is fabricated in volume. A breakdown of the costs is shown in Supplementary Fig. 3 and Supplementary Data 1[26]. The specifications of the device are shown in Table 2.

**Table 2 | Odor delivery device specifications**

| Parameter | Value |
|---|---|
| Number of odor channels | 24 |
| Simultaneous active channels | 3 |
| Maximum airflow rate | 6 L min$^{-1}$ |
| Outlet type | 4-mm pipe |
| Odor reservoir size | 46 mm × 14 mm × 16 mm |
| Noise level | ~60 dB[a] |
| Power consumption | 15 W[a] |
| Digital interface | USB/Bluetooth |
| Dimensions (L × W × H) | 305 mm × 220 mm × 75 mm |
| Weight | 3.9 kg |

[a]In active mode.

## Device characterization

The odorant used to characterize the odor delivery device was developed for the SMELL-RS test (specifically for the SMELL-S subtest) and is described here[12]. For characterization, a volume of 300 μL of the liquid odorant was placed on a cellulose sponge absorber using a micro-pipette and the material was placed in the odor reservoir of the device.

Airflow rates from the outlet channels of the device were measured using a Flusso FLS-110 flow sensor. The mean flow rate and SD were derived from a set of 50 measurement cycles.

Odor intensity was measured using a photoionization detector (MiniPID) from Ion Science. For the measurements of stability, the sensor was positioned at a distance of 25 mm from a 4-mm diameter outlet pipe. Thermal conditions were 25 °C and the odorized airflow rate from the device was 3 L min⁻¹. The mean odor intensity and SD for all PID measurements were derived from a set of ten measurement cycles.

For the short pulse odor intensity repeatability tests, the odor intensity was measured using the PID gas sensor in an indoor environment over a 1 h time window at a temperature of 25 °C, with odor activation times repeated every 300 s.

Thermal stability was assessed by placing the device in an environmental oven (Thermotron S-1.2 3800). The PID gas sensor was mounted externally to the oven, and odorized air was fed to it from the olfactometer using 4-mm diameter Teflon pipes. Prior to each measurement, the system was left to stabilize for 30 min at each temperature point to ensure thermal uniformity.

The spatial distribution of odor intensity was measured by mounting the PID on a motorized stage (Thorlabs, LTS300), having a reach of 300 mm. An image of the test setup is shown in Supplementary Fig. 4. Baffles were placed on either side of the test setup to reduce the effect of air currents which had a greater effect on measured odor intensity variability at longer separation distances.

## Smell test study design

We performed the test-retest reliability and accuracy study with healthy subjects ($n = 37$) and patients with various causes of smell loss ($n = 31$) at Geneva University Hospital. The study was approved by the university hospital's ethics review board, and informed consent was obtained from participants. The study involved subjects aged 18 years of age and over, who came to the hospital for two visits spaced approximately one week apart. During the first visit, participants were tested with the current standard smell test (Sniffin' Sticks) and with SMELL-S with the smell delivery device. We recorded the time needed to complete each test. A Mann–Whitney test was used to uncover differences between groups.

## Sniffin' Sticks smell threshold subtest

Subjects were tested with the Sniffin' Sticks smell test (Burghart, Wedel, Germany), which includes the olfactory threshold, discrimination, and identification sub-tests. The composite score of the three sub-tests was used for the classification of healthy subjects or patients with smell loss[27,28]. The Sniffin' Sticks threshold subtest uses phenylethyl alcohol (rose-like odor) in pen-like odor dispensing devices. The stimuli have 16 dilutions in a geometric series. Three pens were presented in a randomized order, with two containing a solvent and the third the target odorant. The subjects must identify the odor-containing pen. An experimental nurse performed a single-staircase test (with ramped odorant concentrations), with three alternative forced choice procedures starting at the most difficult level (level 16 out of 16) according to the user manual. Reversal of the staircase occurs when the odor is correctly identified in two successive trials. The olfactory threshold was defined as the mean of the last four of seven staircase reversals.

## SMELL-S smell threshold subtest

In contrast to the Sniffin' Sticks threshold test, SMELL-S is self-administered using a computerized app that guides the subjects through the testing paradigm, with subjects entering their responses via the app. The stimulus is composed of a complex odor-mixture, instead of phenylethyl alcohol (rose-like odor) with 10 dilutions in a geometric series. The test starts at a medium difficulty level (level 5 out of 10). The remaining testing procedure is the same as for the Sniffin' Sticks threshold test.

## Statistics and reproducibility

The test-retest reliability and accuracy study included $n = 37$ healthy subjects and $n = 31$ patients with various causes of smell loss. Replicate tests were used to study inter-individual variation with 54 subjects undertaking the Sniffin' Sticks and 67 subjects undertaking the SMELL-S tests, respectively. The order of the tests was randomized, and on the second visit to the hospital, the tests were repeated to check validity (results not shown). A Shapiro–Wilk test was used to test for normal distribution. Because the data was non-parametric, the Mann–Whitney test was used to uncover the timing differences between groups.

## Results

### Device characterization

A number of characterization tests were performed on the odor delivery device to assess the repeatability of the odor delivery, temperature stability, and the spatial distribution of the odor stream. The odorant used for testing was developed for the SMELL-RS test and is described here[12].

Variations in the odorized airflow rate from the device due to poor pneumatic sealing can affect the results of perceptual tests. Initial tests were therefore undertaken with a flow sensor to check the magnitude and uniformity of the airflow. The maximum outflow was measured to be around 6 L min⁻¹. The outflow was adjusted using the regulator valve to provide a mean outflow rate of 3 L min⁻¹ for these tests. The channel-to-channel flow rates are shown in Fig. 2a and show good consistency with a mean outflow rate of 3.0 L min⁻¹ and a maximum deviation from the mean of 5%.

The repeatability of the odor delivery was assessed over an extended time period of operation. To monitor the odor intensity, the odor from the outlet adapter of the device was directed toward a photoionization detector (PID). The PID is extremely sensitive to low levels of organic compounds (<3000 ppb), and signals from the detector yielded a sharp, pulse-like response after the odor activation time, as shown in Fig. 2b, with the odor intensity decaying to background levels after a time period of around 10 s. However, there is likely to be some time lag in the measured results due to the transient response of the detector (~3 s). For the repeatability tests, the delta change in measured odor intensity was extracted from the raw sensor readings.

Normalized PID sensor readings for a number of different short activation times (1–6 s durations) are shown over a 1 h time period in Fig. 2c, measured in indoor conditions. For short odor pulses, the odor intensity is relatively stable, with an RSD of <4.2% for all pulse durations. Temporal variations in odor intensity are likely to be caused by air currents and temperature changes.

Odor intensity measurements were also made with continuous activation of the odor delivery system, and the results of these tests are plotted in Fig. 2d over a 600 s time window. In continuous activation mode, the headspace of the odor reservoir is continually depleted over time, resulting in a decrease in odor intensity. The peak intensity drops by 10% after an activation time of 86 s. If odor intensity stability is required, it is, therefore, important to limit the duration of odor pulses and allow the odor reservoir time to recharge between each activation.

The uniformity of the channel-to-channel odor intensity was also assessed using the PID. Figure 2e shows the PID response for the 24-channels. The measurements show reasonable consistency, with an RSD of 4.7% and a maximum deviation from the mean of 11.4%. Variations in odor intensity between channels could be caused by small differences in the distribution of the odorant on the sponge material, air currents, and temperature changes.

The spatial distribution of the odor stream generated by the device was also investigated. To enable spatial measurements, the PID gas sensor was mounted onto a motorized stage and positioned at various distances away from the odor source, parallel, and across the direction of the odor flow, as

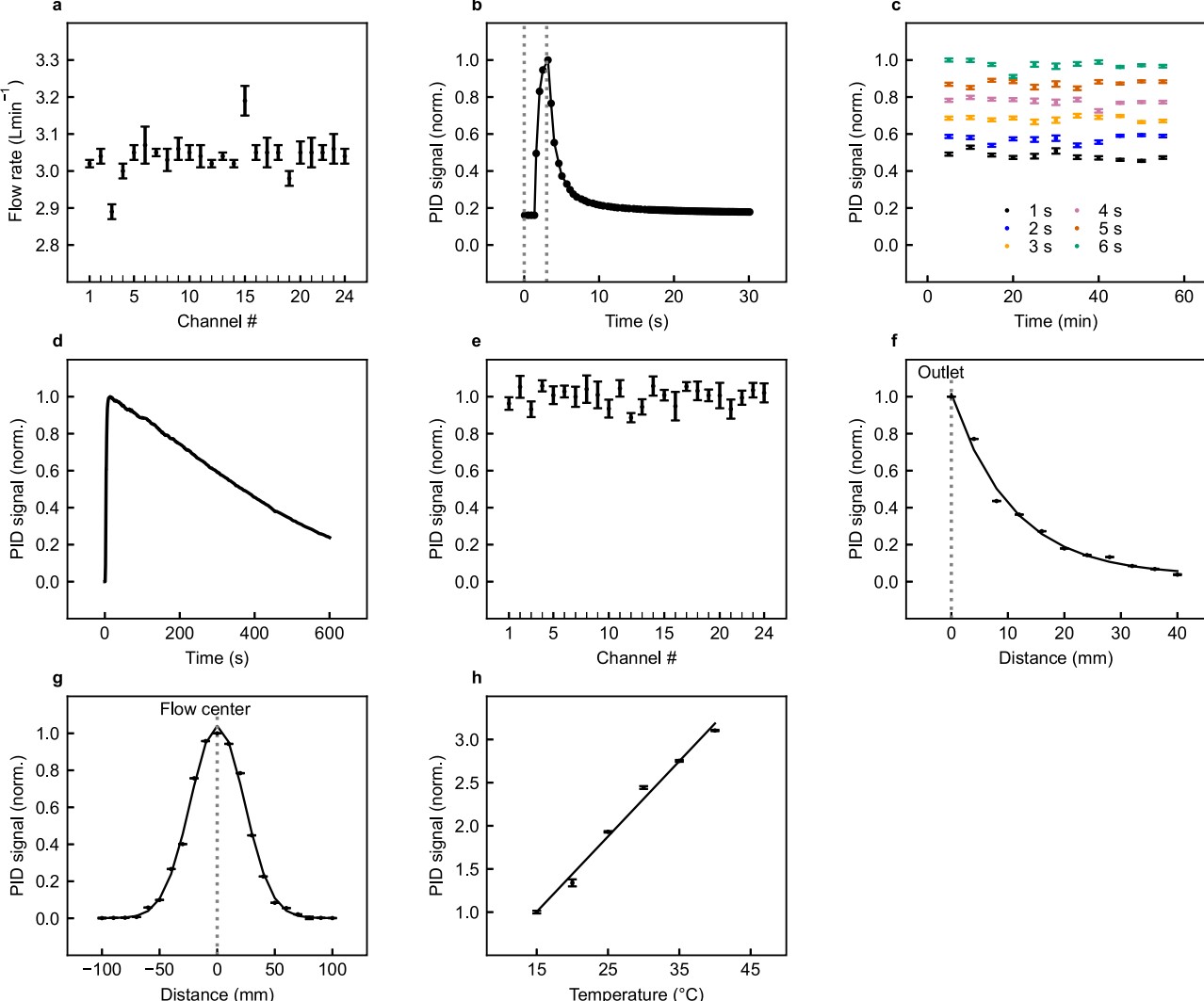

**Fig. 2 | Device performance. a** Measured airflow rates for the 24 outlet channels of the odor delivery device. Mean and standard deviation calculated from a set of $n = 50$ repeat measurements. **b** Transient response of the photoionization detector (PID) to an odor exposure generated by the odor delivery device with a 3 s activation time (activation time indicated by the dotted lines). **c** Temporal stability of the odor intensity generated by the odor delivery device over a 1 h time window with 1, 2, 3, 4, 5, and 6 s activation times. **d** Temporal stability of the odor intensity generated by the odor delivery device over a 600 s (10 min) time window with continuous activation. **e** Channel-to-channel variation in odor intensity generated by the odor delivery device for 3 s activation times. **f** Spatial distribution of odor intensity, measured over a distance of 40 mm from the outlet of the odor delivery device, in the direction of odor flow. **g** Spatial distribution of odor intensity measured across the path of odor flow, at a distance of 100 mm from the outlet of the odor delivery device. **h** Thermal response, showing the variation in measured odor intensity from the odor delivery device with ambient temperature. The mean and standard deviation for all PID measurement data points were calculated from a set of $n = 10$ repeat measurements.

further described in the Methods section. With a simple pipe outlet, the odor intensity drops rapidly in free space, as the odor molecules move and diffuse in all directions away from the outlet, as shown in Fig. 2f. At a 30-mm distance away from the outlet, the odor intensity has dropped to around 10% of the peak value close to the source. The spatial distribution of the odor stream, looking across the airflow at a distance of 10 cm away from the outlet, is shown in Fig. 2g. It is clear from these tests that small changes in the position of the subject under test can have a large effect on perceived odor intensity. To ensure repeatability during smell tests, the subject must, therefore, be accurately aligned with the outlet of the device e.g., by using a headrest.

The temperature stability of the olfactometer system was assessed. For these tests, the device was placed in an environmental oven and the odor intensity was measured using the PID sensor over a range of temperature points (15–40 °C). With the odorant used for these tests, the odor intensity has a measured temperature coefficient of 5% °C$^{-1}$, as shown in Fig. 2h. The temperature stability could be improved by the addition of temperature-controlled odorant reservoirs, at the expense of added cost and complexity.

As the human perception of odor is a logarithmic phenomenon, the effect of temperature-induced changes on perceived odor intensity is less pronounced than one might expect. In addition, the system is intended for use in quasi-thermally stable lab-type conditions.

## Device application for smell testing

To study whether the odor delivery device could decrease the time and human assistance required to administer an olfactory threshold test, we created a customized mobile app to allow for self-administration of the complex testing procedure and used olfactory stimuli from the threshold component of the SMELL-RS concept, called SMELL-S[12].

Olfactory threshold tests use multiple dilutions of an odorant to measure the lowest perceived concentration, analogous to the way that hearing tests measure the lowest perceived intensity of a sound by exposing users to different sound intensities. The SMELL-S test has 10 odorant dilution levels, and the measured odor intensity for each level is shown in Fig. 3a.

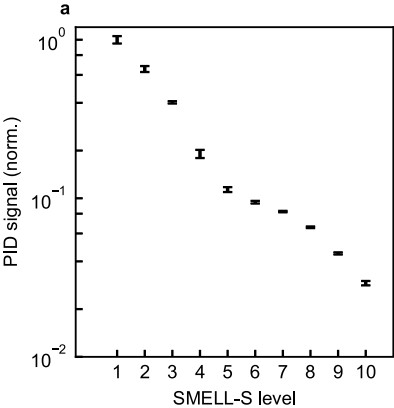
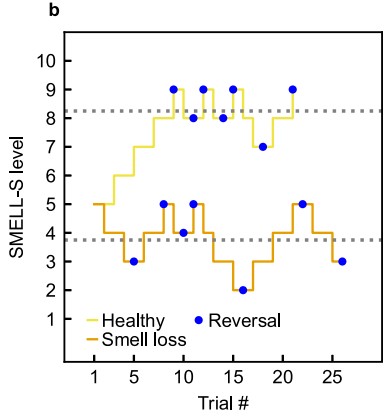
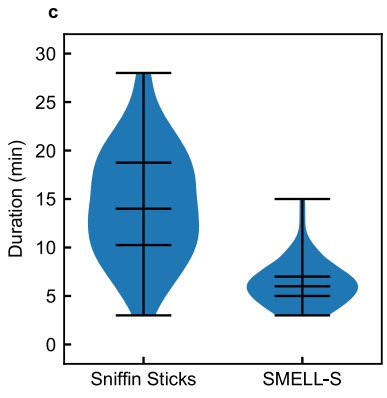

**Fig. 3 | SMELL-S results. a** Odor intensity measured using the photoionization detector (PID) for the SMELL-S odor stimuli. Mean and SD were calculated from a set of $n = 10$ repeat measurements. **b** An example of a subject's performance during the SMELL-S test, for a patient with smell loss treated at the Geneva University Hospital (orange line) and a subject with an intact sense of smell (yellow line). The task becomes more and more difficult as the level approaches 10. A reversal occurs when the direction in which the concentration is changed reverses (blue dots). The score is defined as the mean of the last four reversals (gray dotted lines). **c** Violin plots showing the durations for the Sniffin' Sticks and SMELL-S threshold tests. Lines indicate the median, 25% and 75% quartiles and extrema.

A subject's threshold is tested by performing triangle tests, in which the subject is asked to pick the diluted odorant out of three stimuli that are presented sequentially. Two of the stimuli are solvent controls, and one is the diluted odorant. These triangle tests are used in an adaptive staircase paradigm in which the dilution level is changed throughout the test depending on the subject's performance. Reversal of the staircase is triggered when the odor is correctly identified in two successive trials. Figure 3b shows an example of the performance of two subjects in this paradigm.

We performed a test-retest reliability and accuracy study with healthy subjects and patients with various causes of smell loss and measured the time needed to complete the SMELL-S test with the device, and compared it to the time using the Sniffin' Sticks threshold test (current standard). We found that the median time needed to complete the Sniffin' Sticks threshold test was 14 min (IQR = 9) versus 6 min (IQR = 3) for SMELL-S using the smell delivery device, as shown in Fig. 3c. The interquartile range (IQR) is a measure of variability for non-normal distributions. The two-tailed $p$ value for the Mann–Whitney test with a null hypothesis of no time difference between test types is $p < 0.0001$ for $\alpha = 0.05$. The sum of ranks for Sniffin Sticks and SMELL-S were 4732 and 2650, respectively, and $U = 371.5$.

The time saving when deploying SMELL-S can be explained by the absence of human tasks such as capping/uncapping the Sniffin' Sticks, manual reporting of the subject's answer after each trial, and human interaction between tasks. Such tasks can easily introduce human error, limiting the quality of the clinical data. Although SMELL-S seems to be quicker to administer, it is not yet possible to claim that it will be clinically useful. The goal of the present study is the technical performance of the device, illustrated by two selected clinical cases to show that it may be possible to achieve rapid and accurate smell testing. The clinical validation of SMELL-RS with this device is ongoing. For the moment, we hope that this practical improvement (self-administered, time efficient) will help address, in the near future, an unmet clinical need under the form of a rapid, self-administered, and efficient smell test applicable in different clinical settings around the world.

## Conclusion

We have presented a portable, multi-channel odor delivery device that can deliver a high number of odors flexibly and through personalized digital control. The 24-channel device is more compact and much more affordable compared to existing olfactometer designs. It is self-contained and does not require an external air supply. The use of individual odor channels avoids cross-contamination, and the removable odorant cartridges can be easily exchanged between experimental sessions.

The characterization of the odor delivery device shows that it is possible to deliver multiple odor channels with high temporal precision to users at short distances, making it ideally suited to research and clinical applications, including smell testing.

A comparison with a standard Sniffin' Sticks smell test shows that time savings can be achieved through automation and the removal of human tasks such as capping/uncapping the Sniffin' Sticks. The device's digital integration with an app and cloud-based ecosystem enables efficient data collection from users, removing the need for laborious manual reporting tasks. By modifying the control software and the odors in the cartridges, the device can be used to administer any test that uses 24 or less different olfactory stimuli.

The design provides several unique advantages for investigating smell perception and offers the possibility that users can one day self-administer smell tests in a range of settings, allowing smell healthcare services to evolve and become part of a routine practice of continuous self-monitoring and care for improved health and well-being.

## Data availability

Additional data related to this publication is available from the institutional repository of the University of Cambridge[26].

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

## Acknowledgements

The authors would like to acknowledge the valuable advice of Andreas Keller (The Rockefeller University, New York) for supporting the development of the odor delivery device for the Smell-RS and Smell-S subtest used in this study. We would also like to thank all the healthy subjects and patients with smell disorders for their time and effort in participating in our study and the work of Déborah Blidariu, Katerina Poulopoulou, Dehab Merke, and Sandra Hausmann Jimenez, and the Neurocentre (Nathalie Isidor and Alexie Ray) for supporting the data collection and the management of the SMELL-RS clinical study. Funding for the device development was provided by OWidgets Ltd., and funding for the user study was provided by Geneva University Hospitals. Funding for device characterization was provided by the Department of Engineering at the University of Cambridge.

## Author contributions

E.M. and M.O. conceived the design concept for the odor delivery device. D.P. and R.H. developed the experimental test setup and undertook characterization tests, with F.U. providing technical guidance. B.N.L. and J.W.H. undertook the clinical trials with the device and analysed the results of the trial data. R.H., D.P., and J.W.H. wrote the paper.

## Competing interests

The authors declare the following competing interests: E.M., R.H., and D.P. are employed by OWidgets Ltd. and are working in the field of digital odor delivery technology. M.O. and F.U. act as unpaid advisers for the company. E.M., M.O., and F.U. have >5% share interest in the company. Related patents: Title: "Generating Olfactory Experiences". Description: System and a method of generating olfactory experiences. Application numbers: US 2021/346562 A1 (pending), EP 3784298 A1 (pending); Applicant: University of Sussex. Inventors: E.M. and M.O. Title: "Adaptive Smell Delivery System" Description: A smell delivery device with a delivery channel to produce olfactory output. Application numbers: GB 2600142 A (pending), US 2024/0000368 (pending), EP 4231898 A1 (pending), JP 2023-549141 (pending). Applicant: OWidgets Ltd. Inventors: E.M., R.H., M.O., and F.U.

## Ethical approval

The smell test-retest reliability and accuracy study was approved by the institutional ethics review board of Geneva University Hospital and conducted according to the Declaration of Helsinki on Biomedical Research Involving Human Subjects (IRB approval: 2020-02581).

## Informed consent

Informed consent was obtained from all participants taking part in the study.
