## [Peer Review File · Communications Engineering]

Reviewers' comments:

Reviewer #1 (Remarks to the Author):

The manuscript described a novel device administering olfactory threshold tests. The device constitutes a significant advancement in clinical practice - it is timesaving, easy, compact, multi-channel and does not require external air supply, which is a problem in many clinical settings. I have several comments on the manuscript:

- It is somewhat simplification to say that “a loss of smell usually equates to a loss of taste” please elaborate and explain the perception of aroma through the olfactory system
- IQR abbreviation is unclear, please expand
- By comparing Smell S performed with the proposed device with the Sniffin Sticks threshold test, authors assume that these two olfactory threshold tests otherwise (manually) take the same time. Why haven't the Authors performed Smell-S performed manually with the Smell-S performed with the device? This would be more accurate as Smell-S is already a self-administered method
- I miss the information whether the proposed device can be adapted/programmed to administer other tests, such as the Sniffin Sticks which is the current standard in clinical practice

Reviewer #4 (Remarks to the Author):

OVERVIEW

Olfaction is critically involved in human health, with influence on the ability to assess food quality, socialize, and avoid dangers such as toxins and fire. Moreover, deficits in olfactory perception are heavily implicated with, and a prominent prodromal symptom of, several neurological disorders, including Alzheimer's and Parkinson's diseases. The ability to reliably and efficiently monitor aspects of olfactory perception is thus of tremendous and growing importance to clinical settings. Devices supporting such monitoring are currently limited.

FINDINGS AND SUMMARY

In this study, Hopper et al. describe the design, verification, and validation of a novel odorant delivery device (or “olfactometer”) with potential to overcome several limitations of current devices

and methods for clinically assessing human olfactory perception. In particular, the novel olfactometer presents an intuitive design that provides strong potential for automated, flexible, and efficient delivery of up to 24 stimuli (i.e., different odorants and/or concentrations) suitable for testing different olfactory perceptual qualities (e.g., discrimination and sensitivity), including through self-administration following device setup. The manuscript is well written and data presented are clear and compelling. However, additional verification tests are needed to substantiate several of the design claims, including in particular the ability of the device to reliably and flexibly deliver up to 24 independent stimuli with no cross-contamination. These and other issues to be addressed are enumerated below and organized into major and minor issues.

MAJOR ISSUES

1) The evidence presented clearly demonstrates that the new device is capable of presenting a directed odorant stream with a temporally-controllable pulse. However, no direct evidence is presented to verify that: a) odorant delivery itself from each of the 24 channels is comparable, b) multiple odorants can be effectively presented, c) when multiple odorants are loaded into the cartridge, that minimal cross-contamination across channels occurs, d) “airtight” odorant chambers are, in fact, airtight, e) fast sequencing of odorant delivery is achievable using the mobile app, f) flexible sequencing is achievable using the mobile app. The best evidence supporting these claims, which is fairly indirect, heavily underpowered, and requires quite a leap of faith on the reader’s behalf, is the demonstration of distinct performance of the SMELL-S test using the device by a single healthy and single “Smell loss” patient. More details and verification tests are thus needed throughout. Some possible tests are briefly presented in more specific comments below.

2) Related to the broad point above, comparison of Sniffin Sticks vs. SMELL-S testing durations (Fig. 2h) suggests that the device is able to perform the odor threshold sensitivity task (with different odorant concentrations) similarly to the standard Sniffin’ Sticks approach. However, this is not only somewhat indirect evidence for the above noted points, but this also does not say whether performance outcomes between the Sniffin’ Sticks vs. device were actually comparable. If test duration was shorter with the device but outcomes were markedly different (e.g., if inter-channel cross-contamination reduced subjects’ ability to discriminate between presentations), then it is not at all clear that the device could effectively substitute for the conventional Sniffin’ Sticks.

3) Fig. 2g is also never mentioned in the text. While the authors state that “the clinical performance of SMELL-S will be published elsewhere”, some reference to this figure panel should nevertheless be made, or if not, the panel should be removed. From the methods description, the trial structure in Fig. 2g is also not clear; are trials presented in blocks? If not, why do “Reversal” events sometimes occur at the start of a new SMELL-S level and sometimes in the middle? One would think the Reversal might immediately change the SMELL-S level.

4) Re: comparable delivery from each of the 24 channels: some positional PID sampling across the different tubes would be beneficial to see. For example, how do the positional profiles shown in Fig. 2de change when the same odorant is delivered from the central tube vs. one of the outer and most-angled tubes? Given the fairly sharp drop-offs in PID signal with minor positional shifts shown in Fig. 2de, those small angle differences might lead to measurable differences in concentration and consequent changes in intensity perception, and for repeated patient evaluation, would require loading each test odorant into the same cartridges across each test so as to avoid position-changing confounds.

5) Re: airtight chambers: Can this be demonstrated, either through PID recordings or TiCl₄ smoke visualization?

6) The authors note that “The small headspace of the odour reservoirs allows them to quickly fill with saturated vapour”. This necessarily means that the headspace can also empty quickly, leading to odorant depletion, which may manifest as a drop-off in odorant delivery during long continuous deliveries or a drop-off in delivery peak with rapid repeat deliveries. The documented delivery durations and intervals suggest that these are not major issues (Fig. 2ab). However, to understand the capabilities and limitations of the device (i.e., exactly how fast and flexible is delivery?), and potential other future applications the device might be used for, it would be helpful to: 1) deliver long continuous pulses and use the PID to determine when delivery decrement arises (i.e., repeat Fig. 2a with long odorant pulses), 2) repetitively deliver the same odorant across a range of inter-delivery intervals (i.e., repeat Fig. 2b with shorter intervals), and 3) evaluate the minimal intervals achievable between odorant delivery from two separate channels (i.e., how fast can presentation of different odorants be sequenced).

7) Table 2 lists “Simultaneous active channels” as “3”. Does this mean that up to 3 of the 24 odorant channels can be simultaneously active? This should be clarified. Further description and testing of this potential for simultaneous channel activation and vapor-phase mixture creation should be presented.

8) One potential limitation of the design vs. other olfactometer designs is that there is no un-odorized carrier stream, so there is no potential for controllable vapor-phase dilution at the final delivery point. All dilutions must therefore be performed with liquid-phase mixing when the odorant chambers are prepared. This should be noted, as well as some discussion of whether one of the 24 channels could be converted for such a purpose of generating a continuous carrier/dilution stream. Another consequence of the lack of un-odorized carrier stream is that subjects will go from feeling nothing (pre-odorant delivery) to feeling a puff of air during odorant delivery, so there is mechanotransduction in addition to olfaction occurring. This does not preclude use of the device in

the clinical tests described, but is a key difference compared to other in-lab olfactometer designs that generally take care to ensure a constant air flow is maintained. This difference should thus be noted.

9) PID timecourse plotted in Fig. 2a shows a delay in odor onset from the time of activation (first dotted line). What is this delay? Is it controllable? This may be important for self-administration using a touchscreen (i.e., need some delay in between looking at the phone to tap the start button and redirecting head to the delivery device).

10) Given the sharp drop-off in concentration with parallel distance from the device (Fig. 2d), with PID signals <10% of outlet-measured signals at just 4 cm away from the device, it is somewhat unclear why the authors chose to direct terminal flow tubes to a focal point 10 cm away from the device. Turbulence could render such a focal point somewhat moot, though at the same time the PID recording shown in Fig. 2a is remarkably without turbulent fluctuations (which also merits some discussion). Were alternative resin printed adaptor designs with sharper angles considered to achieve similar odor delivery trajectories across all odor tubes when subjects are positioned closer to the device (i.e., a focal point 3 cm away), thus enabling delivery with less of a drop-off in concentration?

11) Related to the above note on turbulence, visualization of a $TiCl_4$ smoke profile delivered from the device would significantly help the reader understand what the structure of odorant delivery looks like.

12) Are metal odor cartridges reusable? If so, what cross-contamination concerns are there?

MINOR ISSUES

13) Lack of line numbers hinders the review process.

14) References to Appendix figures could be integrated throughout the text to make better use of these figures and further help the reader assess the new device. Currently there are no references to the Appendix.

15) Section 2, Device description, pg. 4: Diaphragm pump has max flowrate of 6 L/min. But downstream of the regulator, “the airflow rate can be adjusted over a range of 2 L/min – 8 L/min”. Max flowrate should be clarified.

16) Terms “odor”, “odorant”, “odour”, and “odourant” are used interchangeably. Terms should be more consistent and intentionally chosen.

17) “2 Device Description” includes the line “This device is developed by OWidgets Ltd., a University spin-out, and of the back of international scientific collaborations”. The phrase “of the back of” seems like a typo; maybe “on the back of” or “with the backing of”

18) Pg. 5: “The odor flow from the pipes is directed toward a focal point 10 cm always from”; “always” should be “away”.

Response to referees

Reviewer #1 (Remarks to the Author):

- It is somewhat simplification to say that “a loss of smell usually equates to a loss of taste” please elaborate and explain the perception of aroma through the olfactory system

We agree with the reviewer’s point and have reworded the text to clarify the possible connection between smell and taste. We have now stated (Line 45) that distortions in olfactory perception usually have a profound impact on the perception of food and drink avors and a reduced ability to enjoy food can have a negative effect on nutrition, and or the immune system.

- IQR abbreviation is unclear, please expand

This abbreviation has been clarified in the text (Line 268).

- By comparing Smell S performed with the proposed device with the Sniffin Sticks threshold test, authors assume that these two olfactory threshold tests otherwise (manually) take the same time. Why haven’t the Authors performed Smell-S performed manually with the Smell-S performed with the device? This would be more accurate as Smell-S is already a self-administered method

We agree that comparing “Manual SMELL-S” vs “Self-administered SMELL-S” would have been a useful and accurate way to show the gain in duration related to the device. However, the Sniffin Sticks threshold test is merely used for comparison because it is currently the most commonly used threshold test in the clinic (gold standard). Furthermore, adding another test (manual SMELL-S) to the study design would have prolonged the duration of the study visit beyond what is tolerable for the study participants. Therefore, we selected the Sniffin’ Sticks threshold test as a comparative group.

- I miss the information whether the proposed device can be adapted/programmed to administer other tests, such as the Sniffin Sticks which is the current standard in clinical practice

By modifying the control software, the device can indeed be adapted for a range of other use cases, aside from the current study. This point has been made in the Conclusion section (Line 354).

Reviewer #4 (Remarks to the Author):

OVERVIEW

Olfaction is critically involved in human health, with influence on the ability to assess food quality, socialize, and avoid dangers such as toxins and fire. Moreover, deficits in olfactory perception are heavily implicated with, and a prominent prodromal symptom of, several neurological disorders, including Alzheimer’s and Parkinson’s diseases. The ability to reliably and efficiently monitor aspects of olfactory perception is thus of tremendous and growing importance to clinical settings. Devices supporting such monitoring are currently limited.

FINDINGS AND SUMMARY

In this study, Hopper et al. describe the design, verification, and validation of a novel odorant delivery device (or “olfactometer”) with potential to overcome several limitations of current devices and methods for clinically assessing human olfactory perception. In particular, the novel olfactometer presents an intuitive design that provides strong potential for automated, flexible, and efficient delivery of up to 24 stimuli (i.e., different odorants and/or concentrations) suitable for

testing different olfactory perceptual qualities (e.g., discrimination and sensitivity), including through self-administration following device setup. The manuscript is well written and data presented are clear and compelling. However, additional verification tests are needed to substantiate several of the design claims, including in particular the ability of the device to reliably and flexibly deliver up to 24 independent stimuli with no cross-contamination. These and other issues to be addressed are enumerated below and organized into major and minor issues.

The specific points raised have been addressed below.

MAJOR ISSUES

1) The evidence presented clearly demonstrates that the new device is capable of presenting a directed odorant stream with a temporally-controllable pulse. However, no direct evidence is presented to verify that: a) odorant delivery itself from each of the 24 channels is comparable,

Tests of the odor intensity generated by the different channels have been made using a photo ionisation detector (PID) to verify the consistency of the odor streams. These results are presented in Fig 2e and described in the text (Line 217). It should be noted that the design of the odor reservoirs for each channel is identical and the flow rates show good consistency, so we would not expect to see big differences in odour intensity between channels.

b) multiple odorants can be effectively presented,

Multiple odour channels were used for the study with different dilutions, however, the mixing of multiple odor streams is not within the scope of the present study, as the work focusses on the application of the device to the SMELL-S test which uses discrete odor streams. Application of the device for odor mixing is the subject of further study.

c) when multiple odorants are loaded into the cartridge, that minimal cross-contamination across channels occurs,

The odor streams use separate paths / channels, greatly minimising the possibility of cross-contamination. The cartridge containing the odorant is constructed from metal and the individual channels are well sealed with O-rings. Cross-contamination would be more likely to occur if there was a common mixing chamber / channel, a detrimental feature which does not exist with the present design. The serial dilution of SMELL-S stimuli correlates with the miniPID results (Fig 3a). With cross-contamination, we would not expect to see this correlation. We have also done tests using a PID sensor and perceptually to assess if adding odorant to an adjacent channel influences an adjacent unodorised channel but have not observed any effect.

d) "airtight" odorant chambers are, in fact, airtight,

To ensure good sealing, the cartridge design incorporates rubber O-rings which form a pneumatic fit under pressure with the metal mating surfaces. To test for sealing, we have measured the air flow rate at the outlet of the different odor channels. During the assembly of devices, we have sometimes recorded differences in the outlet flow rates if channels have not been sealed, as air escapes before reaching the outlet. For the device used in this study, the flow rates recorded for the different channels are consistent with +/- 5 %, suggesting a good quality seal is achieved for all channels. The flowrate test results have been added to the manuscript (see Fig 2a). The term 'airtight' is somewhat subjective so has been removed from the manuscript. In reality, most seals will have some air / odor leakage but we judge the level of air leakage to be negligible for our application. In use, the odourant

cartridge can be easily removed from the device between tests, further limiting the possibility of cross contamination.

e) fast sequencing of odorant delivery is achievable using the mobile app, f) flexible sequencing is achievable using the mobile app. The best evidence supporting these claims, which is fairly indirect, heavily underpowered, and requires quite a leap of faith on the reader's behalf, is the demonstration of distinct performance of the SMELL-S test using the device by a single healthy and single "Smell loss" patient. More details and verification tests are thus needed throughout. Some possible tests are briefly presented in more specific comments below.

We performed a test-retest reliability and accuracy study on a larger number of subjects, including healthy subjects (n = 37) and patients with various causes of smell loss (n = 31) at Geneva University Hospital. Participants were tested with the current standard test (Sniffin' Sticks) and with SMELL-RS using the smell delivery device. These tests were then repeated. The time needed to complete each test and a t-test was used to uncover differences between groups. More details of the user study are presented in the Methods section (Section 5.3).

2) Related to the broad point above, comparison of Sniffin Sticks vs. SMELL-S testing durations (Fig. 2h) suggests that the device is able to perform the odor threshold sensitivity task (with different odorant concentrations) similarly to the standard Sniffin' Sticks approach. However, this is not only somewhat indirect evidence for the above noted points, but this also does not say whether performance outcomes between the Sniffin' Sticks vs. device were actually comparable. If test duration was shorter with the device but outcomes were markedly different (e.g., if inter-channel cross-contamination reduced subjects' ability to discriminate between presentations), then it is not at all clear that the device could effectively substitute for the conventional Sniffin' Sticks.

We completely agree with the reviewer that a shorter test does not mean that it is effective. However, the aim of this paper is not to present data of effectiveness of SMELL-S, which will be presented in a follow-up medical paper, but to highlight the technical aspects of the OW device. The intention in the present paper was not to prove that SMELL-RS could replace Sniffin' Sticks. To make it clearer, we modified the text, and it now reads from Line 273: "Although SMELL-S seems to be faster, it is yet not possible to claim that it will be clinically useful. As a reminder, the goal of the present study is the technical performance of the device illustrated by two selected clinical cases to show that it may be possible to achieve rapid and accurate smell testing. The clinical validation of SMELL-RS with this device is ongoing. For the moment, we hope that this practical improvement (self-administered, time efficient) will help address, in a near future, an unmet clinical need under the form of a rapid, self-administered, and efficient smell test applicable in different clinical settings around the world"

3) Fig. 2g is also never mentioned in the text. While the authors state that "the clinical performance of SMELL-S will be published elsewhere", some reference to this figure panel should nevertheless be made, or if not, the panel should be removed. From the methods description, the trial structure in Fig. 2g is also not clear; are trials presented in blocks? If not, why do "Reversal" events sometimes occur at the start of a new SMELL-S level and sometimes in the middle? One would think the Reversal might immediately change the SMELL-S level.

We have added a description of this figure (now relabelled as Fig 3b) to the text (Line 257). The threshold for a subject is tested by performing triangle tests, in which the subject is asked to pick the diluted odorant out of three stimuli with two stimuli being the solvent controls and one the diluted odorant. An adaptive staircase paradigm is used in which the dilution level is changed throughout

the test depending on the subject's performance. Fig. 3(c) shows the performance of two subjects in this paradigm and is presented as an example.

4) Re: comparable delivery from each of the 24 channels: some positional PID sampling across the different tubes would be beneficial to see. For example, how do the positional profiles shown in Fig. 2de change when the same odorant is delivered from the central tube vs. one of the outer and most-angled tubes? Given the fairly sharp drop-offs in PID signal with minor positional shifts shown in Fig. 2de, those small angle differences might lead to measurable differences in concentration and consequent changes in intensity perception, and for repeated patient evaluation, would require loading each test odorant into the same cartridges across each test so as to avoid position-changing confounds.

To assess variations in the channel-to-channel odor intensity, the odor intensity was recorded with different channel activations and the results are shown in the paper in Fig 2e. Reasonable consistency was achieved in terms of the measured odor intensity channel-to-channel. It should be mentioned that the design of the odor reservoirs is identical and the channel-to-channel airflow rates show good consistency (see Fig 2a).

5) Re: airtight chambers: Can this be demonstrated, either through PID recordings or TiCl₄ smoke visualization?

We have presented the results of flow tests which show the consistency of airflow between the different odor channels, as presented in Fig 2a. Rubber O-rings and the metal construction of the parts provides a good level of sealing.

6) The authors note that "The small headspace of the odour reservoirs allows them to quickly fill with saturated vapour". This necessarily means that the headspace can also empty quickly, leading to odorant depletion, which may manifest as a drop-off in odorant delivery during long continuous deliveries or a drop-off in delivery peak with rapid repeat deliveries. The documented delivery durations and intervals suggest that these are not major issues (Fig. 2ab). However, to understand the capabilities and limitations of the device (i.e., exactly how fast and flexible is delivery?), and potential other future applications the device might be used for, it would be helpful to: 1) deliver long continuous pulses and use the PID to determine when delivery decrement arises (i.e., repeat Fig. 2a with long odorant pulses), 2) repetitively deliver the same odorant across a range of inter-delivery intervals (i.e., repeat Fig. 2b with shorter intervals), and 3) evaluate the minimal intervals achievable between odorant delivery from two separate channels (i.e., how fast can presentation of different odorants be sequenced).

There will be a limit to the odour capacity of our odor headspace system, as the reviewer suggests. The system has been designed with the SMELL-S test in mind which uses short duration (3 s) odor pulses which do not sink a significant volume of odor. To consider the effect of other operating modes, we now present the results of further tests done with longer duration odor pulses and continuous operation (see Fig 2c & Fig 2d). Continuous odor delivery depletes the headspace of the system. This limitation with our headspace system has now been highlighted in the text (Line 210).

7) Table 2 lists "Simultaneous active channels" as "3". Does this mean that up to 3 of the 24 odorant channels can be simultaneously active? This should be clarified. Further description and testing of this potential for simultaneous channel activation and vapor-phase mixture creation should be presented.

Table 2 refers to the simultaneous activation of odor channels. This point has been clarified in Line 159. The number of simultaneous activations is partly limited by the power requirements of the control circuitry. The mixing of different odor types is not something within the scope of the present study, as the work focusses on application of the device with the SMELL-S test which uses multiple discrete odor streams. Application of the device for the mixing of the odor streams is the subject of further study.

8) One potential limitation of the design vs. other olfactometer designs is that there is no un-odorized carrier stream, so there is no potential for controllable vapor-phase dilution at the final delivery point. All dilutions must therefore be performed with liquid-phase mixing when the odorant chambers are prepared. This should be noted, as well as some discussion of whether one of the 24 channels could be converted for such a purpose of generating a continuous carrier/dilution stream. Another consequence of the lack of un-odorized carrier stream is that subjects will go from feeling nothing (pre-odorant delivery) to feeling a puff of air during odorant delivery, so there is mechanotransduction in addition to olfaction occurring. This does not preclude use of the device in the clinical tests described, but is a key difference compared to other in-lab olfactometer designs that generally take care to ensure a constant air flow is maintained. This difference should thus be noted.

The authors agree that the lack of ability to control the dilution level granularly through mass flow control is a limitation of the current design and this point has been highlighted in Line 90. We did investigate the addition of mass flow controllers to vary the flow rate of individual channels but this was felt to increase the cost of the system beyond the level that was acceptable and the device has been developed with the SMELL-S test in mind where odor intensity graduations are achieved by the use of multiple dilutions. An interesting possibility for further research is to mix different odor dilutions to achieve a graduated odor scale.

9) PID timecourse plotted in Fig. 2a shows a delay in odor onset from the time of activation (first dotted line). What is this delay? Is it controllable? This may be important for self-administration using a touchscreen (i.e., need some delay in between looking at the phone to tap the start button and redirecting head to the delivery device).

There is a short delay due to a mixture of factors including activation of the pump (~ 0.5 s), the time it takes the odor stream to reach the sensor and transient response of the sensor. The transient response of the PID sensor is ~ 3 s, so this dominates. This has been clarified in Line 202. The authors agree that it would be useful to establish improved experimental approaches for measuring the transient response time, although this falls outside the scope of this work.

10) Given the sharp drop-off in concentration with parallel distance from the device (Fig. 2d), with PID signals <10% of outlet-measured signals at just 4 cm away from the device, it is somewhat unclear why the authors chose to direct terminal flow tubes to a focal point 10 cm away from the device. Turbulence could render such a focal point somewhat moot, though at the same time the PID recording shown in Fig. 2a is remarkably without turbulent fluctuations (which also merits some discussion). Were alternative resin printed adaptor designs with sharper angles considered to achieve similar odor delivery trajectories across all odor tubes when subjects are positioned closer to the device (i.e., a focal point 3 cm away), thus enabling delivery with less of a drop-off in concentration?

With increasing distance from the outlet, the odor intensity decays in a near inverse exponential manner (as shown in Fig 2f), so at shorter distances there will be more variation in terms of odor intensity than at longer distances. It was thought that a longer distance for the focal point could

therefore have some benefit in terms of reducing odor intensity variations if the position of the user moves. However, we agree that there is a compromise with odor intensity. We have tried steep channel angles to achieve a closer focal point but then the odor streams dissect, meaning that user positioning is even more critical. However, it is true that at longer distances, the effect of external air turbulence is likely to increase, although the user tests were done at close range (10 cm distance). For the extended distance tests, a baffle was used to shield the odor stream, as is now stated in the Methods section. It has been pointed out that the pipes from the device can be connected to different types of outlet adaptor designs, depending on the use case (Line 150).

11) Related to the above note on turbulence, visualization of a $TiCl_4$ smoke profile delivered from the device would significantly help the reader understand what the structure of odorant delivery looks like.

Fig A4 shows in more detail the experimental test setup with the outlet adaptor and sensor. We have not been able to conduct smoke tests but it should be clear from the image how the outlet channels are angled towards the measurement point. The spatial variation in odor intensity is shown in Fig 2f and Fig 2g.

12) Are metal odor cartridges reusable? If so, what cross-contamination concerns are there?

We wanted to make the cartridges reusable / cleanable, so they could be swapped easily and used for different tests. We tested a couple of materials. Initially, the cartridge material was Nylon but this became contaminated due to its porous nature. The cartridge material was then switched to anodised aluminium which can be cleaned effectively using baking soda solution and an oven can also be used to evaporate off any residual odorant. A point about cleaning has been added to the Device description section (Line 141).

MINOR ISSUES

13) Lack of line numbers hinders the review process.

Line numbers have been added to the manuscript.

14) References to Appendix figures could be integrated throughout the text to make better use of these figures and further help the reader assess the new device. Currently there are no references to the Appendix.

References to the appendix figures have been added throughout the text.

15) Section 2, Device description, pg. 4: Diaphragm pump has max flowrate of 6 L/min. But downstream of the regulator, "the airflow rate can be adjusted over a range of 2 L/min – 8 L/min". Max flowrate should be clarified.

The stated flowrate has been corrected. The maximum flow rate of the device is 6 L/min. The pump can deliver a higher flow rate but the effect of back pressure reduces this slightly.

16) Terms "odor", "odorant", "odour", and "odourant" are used interchangeably. Terms should be more consistent and intentionally chosen.

The terms 'odor' (for the gaseous phase) and 'odorant' (liquid phase) have now been used throughout the manuscript.

17) "2 Device Description" includes the line "This device is developed by OWidgets Ltd., a University spin-out, and of the back of international scientific collaborations". The phrase "of the back of" seems like a typo; maybe "on the back of" or "with the backing of"

18) Pg. 5: "The odor flow from the pipes is directed toward a focal point 10 cm always from"; "always" should be "away".

These typos have been corrected.

REVIEWERS' COMMENTS:

Reviewer #4 (Remarks to the Author):

Revisions made by the authors, in particular the addition of new PID recordings as well as added description of the odorant chambers/cartridge, have satisfactorily addressed my previous comments. The manuscript now provides a more comprehensive account of the novel device and more concretely supports the design claims. As noted by Reviewer #1, some confusion/errors may still surround the IQR values listed on line 267: Fig 3c shows larger IQR for Sniffin Sticks than SMELL-S, yet the text reports the opposite. This should be easily addressed without further review.

Reviewer #4 (Remarks to the Author):

Revisions made by the authors, in particular the addition of new PID recordings as well as added description of the odorant chambers/cartridge, have satisfactorily addressed my previous comments. The manuscript now provides a more comprehensive account of the novel device and more concretely supports the design claims. As noted by Reviewer #1, some confusion/errors may still surround the IQR values listed on line 267: Fig 3c shows larger IQR for Sniffin Sticks than SMELL-S, yet the text reports the opposite. This should be easily addressed without further review.

We have corrected the IQR values in the fourth paragraph in the Results 'Device application to smell testing' subsection.